Augmented reality in healthcare education: an integrative review

Zhu Egui 1 2
Hadadgar Arash 1
Masiello Italo 1
Zary Nabil 1 Nabil.Zary@ki.se
1 Department of Learning, Informatics, Management and Ethics (LIME), Karolinska Institutet , Stockholm , Sweden
2 Faculty of Education, Hubei University , China
Hochheiser Harry
Electronic publication date: 2014 Jul 8
Publication date: 2014
Volume: 2
Electronic Location ID: e469
Received 2014 Apr 1; Accepted 2014 Jun 16
Copyright: © 2014 Zhu et al.
Copyright year: 2014
Copyright holder: Zhu et al.
License: This is an open access article distributed under the terms of the Creative Commons Attribution License, which permits unrestricted use, distribution, reproduction and adaptation in any medium and for any purpose provided that it is properly attributed. For attribution, the original author(s), title, publication source (PeerJ) and either DOI or URL of the article must be cited.
License URL: https://creativecommons.org/licenses/by/4.0/

Keywords: Medical education, Medical simulation, Augmented reality

Funding: Karolinska Institutet The project was funded with intramural funds from Karolinska Institutet. The funders had no role in study design, data collection and analysis, decision to publish, or preparation of the manuscript.

==============================
Background. The effective development of healthcare competencies poses great educational challenges. A possible approach to provide learning opportunities is the use of augmented reality (AR) where virtual learning experiences can be embedded in a real physical context. The aim of this study was to provide a comprehensive overview of the current state of the art in terms of user acceptance, the AR applications developed and the effect of AR on the development of competencies in healthcare.

Methods. We conducted an integrative review. Integrative reviews are the broadest type of research review methods allowing for the inclusion of various research designs to more fully understand a phenomenon of concern. Our review included multi-disciplinary research publications in English reported until 2012.

Results. 2529 research papers were found from ERIC, CINAHL, Medline, PubMed, Web of Science and Springer-link. Three qualitative, 20 quantitative and 2 mixed studies were included. Using a thematic analysis, we’ve described three aspects related to the research, technology and education. This study showed that AR was applied in a wide range of topics in healthcare education. Furthermore acceptance for AR as a learning technology was reported among the learners and its potential for improving different types of competencies.

Discussion. AR is still considered as a novelty in the literature. Most of the studies reported early prototypes. Also the designed AR applications lacked an explicit pedagogical theoretical framework. Finally the learning strategies adopted were of the traditional style ‘see one, do one and teach one’ and do not integrate clinical competencies to ensure patients’ safety.

Introduction

Augmented reality (AR) supplements the real world with virtual objects, such that virtual objects appear to coexist in the same space as the real world (Zhou, Duh & Billinghurst, 2008). It has the potential to provide powerful, contextual, and situated learning experiences, as well as to aid exploration of the complex interconnections seen in information in the real world. Students can use AR to construct new understanding based upon their interactions with virtual objects, which bring underlying data to life. AR is being applied across disciplines in higher education, including; environmental sciences, ecosystems, language, chemistry, geography and history (Johnson et al., 2011; Klopfer & Squire, 2007). Clinical care is also interested in AR because it provides doctors with an internal view of the patient, without the need for invasive procedures (Bajura, Fuchs & Ohbuchi, 1992; Chris, 2010; De Paolis et al., 2011; De Paolis et al., 2008; Pandya, Siadat & Auner, 2005). Since students and medical professionals need more situational experiences in clinical care, especially for the sake of patient safety, there is a clear need to further study the use of AR in healthcare education. The wide interest in studying AR over recent years (Rolland et al., 2003; Sielhorst et al., 2004; Thomas, John & Delieu, 2010) has highlighted the following beliefs:

• AR provides rich contextual learning for medical students to aid in achieving core competencies, such as decision making, effective teamwork and creative adaptation of global resources towards addressing local priorities (Frenk, Chen & Bhutta, 2010).

• AR provides opportunities for more authentic learning and appeals to multiple learning styles, providing students a more personalized and explorative learning experience.

• The patients’ safety is safeguarded if mistakes are made during skills training with AR.

While information technology has been presented as a driver for educational reforms, technology, by itself, is not a vehicle for learning (Merrill, 2002; Salomon, 2002). To prevent AR from being a gimmick with tremendous potential, it is important to understand the new capabilities that technology offers, including how those capabilities may impact learning (Garrison & Zehra, 2009; Salinas, 2008). Therefore, a necessary first step is to analyze the current research on AR in healthcare education to determine its’ strengths and weaknesses.

There are two systematic reviews about AR; one is on AR in rehabilitation to improve physical outcomes (Al-Issa, Regenbrecht & Hale , 2012), and the other is focused on AR tracking techniques (Rabbi, Ullah & Khan, 2012). In addition to these, Lee (2012) published a literature review to describe AR applied in training and education, and discussed its potential impact on the future of education. Carmigniani & Furht (2011) developed an overview of AR technologies and their applications to different areas. Shuhaiber (2004) discussed augmented reality in the field of surgery, including its potential in education, surgeon training and patient treatment. Thomas, John & Delieu (2010), provided a brief overview of AR for use in e-health within medicine, and specifically highlighted issues of user-centered development. Ong et al. (2011) presented the use of AR in assistive technology and rehabilitation engineering, focusing on the methods and application aspects. Of the studies include in these reviews, only two focused on medical or healthcare education. The first reviewed the current state of mixed reality manikins for medical education (Sherstyuk et al., 2011). The second analyzed applying AR in laparoscopic surgery with a focus on training (Botden & Jakimowicz, 2009). Both were helpful in understanding AR from different perspectives, but lacked a broader view of AR in healthcare education. Furthermore, few reviews focused on analyzing AR in relation to learning and teaching, which is important for ensuring that AR has an appropriate instructional design adapted to medical education.

The aim of this study was therefore to investigate of the current state of AR in healthcare education and its reported strengths and weaknesses of the reported AR applications for education in healthcare.

Material and Methods

We conducted an integrative review, as described by Whittemore (2005). Integrative reviews are the broadest type of research review method and allow for the inclusion of various research designs to more fully understand a phenomenon of interest. In contrast, systematic reviews combine the evidence of primary studies, with similar research designs, to study related or identical hypotheses (Whittemore, 2005). They are more useful at providing insights about effectiveness, rather than seeking answers to more complex search questions (Grant & Booth, 2009). Scoping reviews identify the nature and extent of research evidence to provide a preliminary assessment of the potential size and scope of available research (Grant & Booth, 2009). However, due to lack of a process for quality assessment, the findings from scoping reviews are weak resources for recommending policy/practice. Performing an integrative review helped us to understand how AR has been applied in healthcare education, and our findings will be used to help guide better practice. Our review included multi-disciplinary research publications, reported until 2012, which were related to the construct of AR in healthcare education.

Inclusion and exclusion criteria

In order to present a comprehensive overview of AR relative to healthcare education, we used broad inclusion criteria. An article was included in the review if it was a research paper (*), on AR (**), or on AR in healthcare education (***), and was written in English. The criteria of inclusion and exclusion were further defined as follows in Table 1:

Table 1 The inclusion and exclusion criteria.

Criterion	Inclusion criteria	Exclusion criteria	
Research	• Clearly described the goal or research question	• Neither goal nor research question described	
	• A scientific study design	• Review papers were put in introduction	
	• The data collection and analysis methods were clearly described		
	• The results were clearly described		
Focus of the technology	• Combination of real and virtual environments	• Used mixed reality in name, but was only virtual reality.	
	• Interactive in real-time		
	• Real or perceived registration in 3D		
Content	• Healthcare education	• Education without medicine	
	• Health science education	• Medicine without education	
	• Medical education	• Veterinary medicine education	

Clarification of criteria terms:

• Research paper. There is no widespread accepted set of criteria with which to assess the quality of studies. Further, research paradigm is different across the various members of the academic community, such as developer, educator and doctor. We have not restricted the methodology and the writing style of the research papers but they should contain the following core information; clear description of the context, study aims, research question, study design, sampling, data collection and analysis, and findings. Papers were excluded if they did not describe the core information mentioned above.

• AR. Augmented reality, which sometimes is referred to as ‘mixed reality‘, or ‘blended reality,’ is a technology that allows a live real-time direct or indirect real-world environment to be augmented/enhanced by computer-generated virtual imagery information (Carmigniani & Furht, 2011; Lee, 2012). It is different from virtual reality that completely immerses the user in a computer-generated virtual environment. We did not make a clear distinction between augmented reality and augmented virtuality (AV) where AR is closer to the real world and AV is closer to a pure virtual environment (Milgram & Colquhoun, 1999). Studies focusing on enhancing the user’s perception of and interaction with the real world through virtual information were included. It would be excluded if it only discussed the virtual environment.

• Healthcare education. According to the glossary of medical education terms from AMEE, medical education is “the process of teaching, learning and training of students with an ongoing integration of knowledge, experience, skills, qualities, responsibility and values which qualify an individual to practice medicine” (Wojtczak, 2002, p 36). “With the growing understanding of the conditions for learning within medical care and health care, and the increasing focus on the ‘lifelong’ nature of medical education, medical education now, more so than in the past, needs to span three sectors: undergraduate, postgraduate and the continuing professional development of established clinicians” (Swanwick & Buckley, 2010, p 123). The two definitions represents to current established perspectives on medical education, the first with a process and outcome focus, while the second is acknowledging education as a lifelong continuum.

Search strategy and inclusion procedure

Agreement about the review protocol, and inclusion and exclusion criteria was reached through discussion between EZ, NZ and IM. Relevant computerized databases were searched for eligible studies, including: ERIC, CINAHL, Medline, Web of Science, PubMed and Springer-link. Separate searches were completed for each database with no date restrictions, no methodological filter, and the language limited to English. The searches were updated until November 2012. Word groups representing the key characteristics of our study were created and combined in several ways. The first group was ‘augmented reality’ and included terminology with similar meaning such as ‘mixed reality,’ or ‘blended reality.’ The second group was ‘medical education’ and included terms like ‘healthcare education,’ ‘health science education’ and so on. The two key groups of terms used the Boolean operator ‘(and)’ to combine with the terms one another when searching for papers to include. Also, we used symbols like ‘medic * education’ to include more related articles with potentially different endings.

EZ independently searched for eligible studies in the six databases using the methods above and identified each article meeting the inclusion criteria. ‘Medical education’ and synonyms were searched in ‘all areas’ in the six databases throughout the search procedure. We began by searching for ‘augmented reality,’ or synonyms, plus ‘medical education’ in all areas to get the overall data. Next, ‘augmented reality,’ or its’ synonyms, were searched within the title or abstract field, but with ‘medical education’ in ‘all areas.’ One reason for this is that we felt the focus terms should be placed in the title or abstract. Another reason is that the papers in which augmented reality was neither in the title nor abstract, were not studying augmented reality when we reviewed them. When the abstract contained insufficient information we sometimes referred to the full text to assess eligibility. This was then discussed with NZ and IM. After confirming that the paper’s title and abstract discussed augmented reality on medical education, the full text was downloaded and printed to re-read and analyze, if it met the review criteria. EZ examined and marked the full texts to select the articles that met the inclusion criteria. AH checked the excluded papers by EZ to ensure we did not leave out any papers that should include. NZ checked the full text and discussed with EZ. IM was involved in the discussions and selection process when necessary. The quality of the studies was then reviewed by all the co-authors for final inclusion.

Data extraction and analysis

We extracted information specifically on research, technology and learning from the included studies. The characteristics and the results of the included studies were recorded with a standardized data-extraction form. Data were extracted independently and in tripartite for all characteristics. Three main characteristics, including research, technology and learning, and eleven sub-characteristics were described through qualitative content analysis for each of the included studies (Appendix SI). Also, we used content analysis to describe the study design for each study (Appendix SI). Thematic analysis was used to identify the prominent themes that describe current use of AR in healthcare education. The themes are then presented in the result section in terms of strength and weakness of AR.

Results

Identification of relevant studies

We found 2,529 papers on AR in medical education in the above-mentioned six databases. After screening the titles and abstracts, we found 270 citations in the titles and 179 in the abstracts that included ‘augmented reality’, ‘mixed reality’ or ‘blended reality.’ These terms were selected to keep focus on the key characteristics that we wanted to scrutinize and identify. After further reading of the title and abstracts, and removal of any duplicate papers, 77 full-text papers were retrieved and reviewed in more detail. Twenty-five articles met our inclusion criteria for data extraction and were analyzed. Figure 1 shows the selection process. Papers were mainly excluded if their research aim and context were not clearly described. Some articles which seemed to discuss medical education were later excluded because they only focused on medicine or treatment, and not on healthcare education (Bruellmann et al., 2012; Di Loreto et al., 2011; Pagador et al., 2011), and vice versa, one was excluded because it discussed education of another discipline that could contribute to the health of students (Hsiao, 2012).

Figure 1 Literature search and selection flow.

From the included 25 research papers focusing on AR in healthcare education, 20 were based on quantitative research methods, 3 on qualitative research methods and 2 on mixed research methods. In these studies, AR was applied on 15 healthcare related subjects. Most of studies used their own AR system and 5 groups used the same system.

Methodological quality of the identified studies

We chose to apply a broad inclusion criteria and no restriction with regard to the papers’ methodology since research on AR is still in an early innovative phase. Methodological quality was presented adapting the Medical Education Research Study Quality Instrument (MERSQI) (Reed et al., 2007). Quality (Table 2) was assessed purely for descriptive purposes, not as grounds to exclude.

Table 2 Characteristics of the included studies.

Characteristics	Types	No of studies	
Study design	Experiment	1 group post-test only	2	
		1 group pre-test and post-test	1	
		2 groups randomized	6	
		2 groups non-random	5	
		3 groups non-random	4	
	Descriptive	Interviews	3	
		Questionnaire	10	
		Case	2	
Type of data	Self-reported (participants)	10	
	Measured	18	
Data analysis	Descriptive analysis	19	
	Other types of analysis	3	
Outcomes	Satisfaction, attitudes, perceptions, opinions	10	
	Knowledge, skills	16	
	Experiences	2	
	Healthcare outcome	0	
	Not reported	3	

Use of augmented reality in healthcare education

The earliest study on AR in healthcare education was published in 2002 but publications in the field take off starting in 2008 (see Appendix SI). Fig. 2 was developed to map our results on AR in healthcare education, and to give us a clearer understanding of learning paradigms and the capabilities of AR offered in current research.

Across the studies we saw high variability in the research aims and also the role of AR in healthcare education (Fig. 2). Twelve studies focused on evidence that AR can improve learning. Seven studies were aimed at developing AR systems for healthcare education. Two studies investigated the user’s acceptance of AR as a learning technology. Six studies tested AR applications. The main use of AR for learning has been to provide feedback, and eight studies used AR as a means to provide feedback to students. Two studies used AR as an innovative interface and two studies used it for simulator practice. The other studies tried AR as navigation, regenerative concept, remote assessment and training, and as a meaningful information tool. One used it to reduce resources, while another group used it to offer immersion in a scenario, and one tried to give participatory reality.

The research results showed that learners can accept AR as a learning technology, and that AR can improve the learning effect by acquisition of skills and knowledge, understanding of spatial relationships and medical concepts, enhancing learning retention and performance on cognitive-psychomotor tasks, providing material in a convenient and timely manner that shortens the learning curve, giving subjective attractiveness, and simulating authentic experiences (see Appendix SI and Fig. 2).

Figure 2 Characteristics of AR in medical education.

* This number is the total of unique participants for all the included papers. We used the largest number given for two groups (Botden et al., 2007; Botden et al., 2008; Botden, Hingh & Jakimowicz, 2009a; Botden, Hingh & Jakimowicz, 2009b; Leblanc et al., 2010a; Leblanc et al., 2010b; Leblanc et al., 2010c; Leblanc et al., 2010d), who published 4 papers; ** This number shows the type of computer system that was used in the included papers. Three papers did not describe a computer system (Karthikeyan et al., 2012; Sakellariou et al., 2009; Yudkowsky et al., 2012).

Technical specifications

Most of the included papers (50%) employed mobile laptops. Four studies used smaller mobile devices such as smart phone, tablet, PDA and e-book readers. Seven papers used stationary desktop computers. Three papers did not mention which computing system they used in their studies.

Of the included papers, 68% used a camera and marker as a tracking device. Two papers used an electromagnetic tracker but different markers; one a radiographic marker and one used anatomical landmarks. Two papers used sensors. Other tracking systems, such as hybrid optical tracker and Wi-Fi signal, were found in at least one of the included papers. One paper described using a head-and-hand tracking system, but did not provide details on the technology (Yudkowsky et al., 2012). One did not use a tracking device because they projected the virtual picture on a manikin (Pretto et al., 2009).

Strengths of AR in healthcare education

We identified three themes that related to the strengths of AR in healthcare education.

AR implemented in several healthcare areas and aimed at all level of learners

AR was applied in various subjects, such as: joint injection, thoracic pedicle screw placement, laparoscopic surgery, administering local anesthesia, endotracheal intubation, ventriculostomy, forensic medicine, inguinal canal anatomy, diathermy, tissue engineering, alimentary canal physiology and anatomy, disease outbreak, clinical breast examination, cardiologic data, and life support training, all of which are applicable to healthcare education (see Fig. 2). We found that 64% of the included papers were within surgery, primarily laparoscopic surgery, which represented 44% (11/25). Two groups provided the majority of publications of laparoscopic surgery (Botden et al., 2007; Botden et al., 2008; Botden, Hingh & Jakimowicz, 2009a; Botden, Hingh & Jakimowicz, 2009b; Leblanc et al., 2010a; Leblanc et al., 2010b; Leblanc et al., 2010c; Leblanc et al., 2010d. Other healthcare subject areas had only one paper included in this research.

While two studies did not mention participants, the remaining 23 studies included 713 participants representing medical staff, medical students, high school students and children, (see Table 2 and Fig. 2). Participants used AR to learn healthcare skills and aquire knowledge. Most of the participants were, or will be, healthcare staff, however the children and high school student participants may not pursue an education or career in healthcare in their future.

AR seems useful for improving healthcare education

Ninety-six percent of the papers claimed that AR is useful for improving healthcare education. Several aspects were elicited in the different studies such as decreased amount of practice needed, reduced failure rate, improved performance accuracy, accelerated learning, shortened learning curve, easier to capture learner’s attention, better understanding of spatial relationships, provided experiences with new kinds of authentic science inquiry and improved assessment of trainees.

Broad focus of research—from user acceptance, system development and testing, to the study of learning effects

Even though every paper in this study had its own research aim and focus, together they gave us a more complete perspective of how AR is being used in healthcare education (Fig. 2). Two papers investigated user acceptance of AR and they claimed that participants would like to use AR instructions in their future professional life, primarily due to the perceived usefulness of AR. Six papers focused on developing AR systems and two of them tested the usefulness of the systems. One of the six studies, in addition to two other studies focused on evaluating the validity of AR systems. One paper described the usefulness, reliability and applicability of the AR system, and one tested the system value. Fourteen out of the twenty-five papers presented AR for various learning aims.

Weaknesses of AR in healthcare education

We also identified three themes around the weaknesses of AR in healthcare education.

Lack of learning theories to guide the design of AR

Of the included papers, 80% did not clearly describe which kind of learning theory was used to guide design or application of AR in healthcare education. One claimed that they used activity-based learning but did not tell us how they used it; moreover, the learning strategies are not clearly described in the paper (Sakellariou et al., 2009). Two groups used standard skills, such as the manual skills of fundamentals of laparoscopic surgery or expert illustration of what is done in practice, to guide design of AR systems. The participants in these groups used the standard skills identified to perform a task. One group, which used situated learning, allowed the participants to explore and navigate with AR environments, but did not show any learning effect (Rasimah, 2011). Only one group used on location learning theory and the learning strategy of collaborative inquiry and role play (Rosenbaum, Klopfer & Perry, 2007). The results indicated that incorporating the affordances of AR games and the dynamic models of participatory simulations make possible new kinds of authentic science inquiry experiences.

Traditional learning strategies applied

In 64% of the included papers it was shown that they are still using traditional methods of teaching practical skills in medical education, whether or not AR was used as a guidance system or as feedback tool. Three included papers (12%) did not describe how the participants used AR to learn. One wrote that students can explore and navigate with AR environments, but that the time allotted was only half an hour and no learning effect was shown (Rasimah, 2011).

However, a few studies explored other methods. One study investigated AR in teaching using different forms such as; group setting, self-learning or revision of cases (Jan, Noll & Albrecht, 2012). One research group used interactive story and another group used game play to attract students (Karthikeyan et al., 2012; Nischelwitzer et al., 2007). One group used collaborative inquiry and role-play strategies (Rosenbaum, Klopfer & Perry, 2007).

Mostly AR applications prototypes reported

Fifty-six percent of the papers presented an AR prototype without studying its impact. Five groups studied the ProMIS AR simulator, which was used by colorectal surgeons in their training to improve laparoscopic colorectal skills. The 5 groups contributed with 11 papers. The usefulness, reliability and applicability of the ProMIS AR simulator system were examined, and the systems’ value and validity were also evaluated. ProMIS AR was additionally compared to other systems.

Discussion

In this paper, we have shown an overview of the use of AR in healthcare education, additionally, we have identified the currently reported strength and weakness. The findings suggests a potential role in healthcare education even if most of the AR applications were still in a prototype stage.

Most studies said AR is useful for healthcare education, with one exception that did not mention the learning effect of AR. AR is useful because it helps the healthcare learner to understand spatial relationships and concepts, to acquire skills and knowledge, to strengthen cognitive-psychomotor abilities, and to shorten their learning curve and prolong learning retention. Further, it increases subjective attractiveness by providing students with authentic simulated experiences. Moreover, AR offers more conveniences, such as with time.

Most of the studies used AR for learning through feedback or as a navigation system. However, a few used AR to offer immersion into a scenario, a participatory reality or a regenerative concept. Some used AR as an innovative interface or meaningful information tool. The others tried AR for remote assessment and training, or simulator practice. One used it to reduce resources.

Comparison with existing literature

Two pieces of literature relevant to healthcare education, focused on introducing several examples of using AR systems. Sherstyuk et al. (2011) introduced human manikins with augmented sensory input for medical education, while Botden & Jakimowicz (2009) compared three AR systems that allow the trainee to use the same instruments currently being used in the operating room for laparoscopic surgery. Al-Issa, Regenbrecht & Hale (2012) used systematic review to investigate the effectiveness of physical outcomes through use of AR in rehabilitation. AR is not currently included in rehabilitation training and the study also showed that research on AR in rehabilitation is still in its infancy. Rabbi, Ullah & Khan (2012) attempted a systematic review of AR tracking techniques but did not show a result. Carmigniani & Furht (2011) focused on analysis of the technical specifications of different types of AR and pointed out the advantages and disadvantages. They also discussed AR for use in medicine and education.

This review searched six different databases to determine the characteristics of AR in healthcare education, and to distinguish the strength and weakness found in current research. It particularly focused on including studies related to healthcare education. Most of the AR applications found in this review are based on mobile computing systems, especially on laptops. It is different with Carmigniani and Furht’s study where the medical AR application systems are fixed in-doors. While light mobile AR has been predicted to be feasible to develop as real-time AR applications that are locally processed, our findings show that there are still very few examples of light mobile AR (Carmigniani & Furht, 2011). In our review, we aimed to not only describe the research outcome and learning effect of included papers, but also to check which kind of learning theory was used and how they used it.

AR and educational theory

Although each study presented a clear research aim, few suggestions were given for choosing an AR model that is better for healthcare education. Moreover, there is not enough evidence to inform the design of suitable learning activities with AR system, where knowledge and skill development could be integrated into the learner’s world. Thus, further research in this area should be taken to clarify the appropriate AR model, instructional designs and how to effectively use AR for healthcare education.

Study strengths and limitations

To our knowledge, this is the first integrative review that specifically addresses AR in healthcare relative to education. We explored how AR was applied in healthcare education encompassing a broad range of learners, learning strategy, outcomes and study designs. Content analysis and thematic analysis were useful to provide a comprehensive understanding on AR in healthcare education.

This review tries to provide a comprehensive description of AR in healthcare education with no research methodology filter. However, it is possible that some studies were missed if the key words did not appear on the title or abstract. The studies were also limited by excluding any non-English studies. This was not only because most of papers were published in English, but also because the authors come from different countries, and English allowed them to reach a consensus on the articles to include in the analysis. It is useful to minimize bias, but we possibly excluded some important papers. Further, an interesting AR application could have been missed because it was not published.

Conclusions

AR is in the early stages of application within healthcare education but it has enormous potential for promoting learning in healthcare based on this review of preliminary AR studies. The infancy of AR in healthcare education requires more than the testing and improvement of prototype products, but also needs to identify appropriate learning theories to better guide application of AR in healthcare education.

Supplemental Information

Appendix SI Appendix I: Description of 25 comparative studies included in the integrative review of AR in medical education

Click here for additional data file.

We wish to thank Anneliese Lilienthal for her role in reviewing this manuscript as a native English speaker, and helping to improve the quality of the language presented here.

Additional Information and Declarations

Competing Interests

Author Contributions

The authors declare there are no competing interests.

Egui Zhu conceived and designed the experiments, performed the experiments, analyzed the data, wrote the paper, prepared figures and/or tables, reviewed drafts of the paper.

Arash Hadadgar and Italo Masiello analyzed the data, reviewed drafts of the paper.

Nabil Zary conceived and designed the experiments, performed the experiments, analyzed the data, wrote the paper, reviewed drafts of the paper.

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
