# Peer review of "Augmented reality in healthcare education: an integrative review"

_PeerJ, doi:10.7717/peerj.469_

## Round 0.1 · original submission · Major Revisions

1. Reviewer 1 expresses concern that this is a literature review, and therefore out of scope for PeerJ. I have discussed this question with PeerJ staff and they have indicated that this paper is in spirit close to a systematic review, and therefore within scope for PeerJ.

2.Reviewer 2 provides some constructive criticisms that are quite appropriate, particularly with respect to
spelling and grammar (which need significant work), the moving of Table 3 to an appendix, an explanation of the goals and methods, and the conclusions.

3. Elaborating on Reviewer 2, I felt that the notion of "integrative" review could use a bit of discussion, particularly with respect to differentiating this type of review from both systematic reviews and scoping reviews. I think this paper would be stronger if more detail on methods (precise search terms and criteria) were spelled out more explicitly and if the review could synthesize a bit more discussion about the state of the research on this topic and promising areas for further investigation.

·

Basic reporting

Some unclear and awkward phrasing occurs occasionally in the text.

Experimental design

I believe this manuscript is best defined as a literature review. While there are some worthwhile conclusions produced by the review, it does not appear to be original primary research. No original data were presented.

Validity of the findings

The "findings" as they are, seem sensible as they logically follow from the review, but no data are present.

Additional comments

I thought the topic, breadth, and conclusions from the literature review to be very sensible. From what I can tell, this manuscript does not fall within the scope of what PeerJ publishes. I think there may be other more suitable places for this paper. I suggest having someone proofread the article as well--there are grammatical problems and some typos throughout.

Reviewer 2 ·

Basic reporting

- The article should be shortened considerably, and a spell & grammar check should be done.
- Table 3 may be included in an appendix.
- An explanation why the study was done should be given in more detail, as well as
a clear conclusion should be provided.

Experimental design

The description of the literature search and the selection criteria are described
in sufficient detail.

Validity of the findings

The conclusion is only partly based on the data that have been generated.
The fact that AR is at the early stage in healthcare education is not really astonishing and the statement that it has "enormous potential" would need more specific arguments. It may be interesting to study in more detail potential hurdles and give more guidance for thos working in this or related fields.

---

## Round 0.2 · Minor Revisions

Reviewer 1 lists a number of minor points that can be addressed relatively easily through textual edits and localized revisions. Please address these comments to the extent possible. Some - such as the comment about the need for updating of the analysis might simply be addressed by a discussion point acknowledging the evolution of the the field.

I believe that this paper will be suitable for publication if these changes are made.

·

Basic reporting

This is a rewritten submission. I can see where the authors have edited their work based on prior reviewer feedback and I thank them for their attention. Issues I found:

Lines 88-91: 3 goals (user acceptance, applications, healthcare competencies) are mentioned but I don't see repeated acknowledgement of these later on. It would be wise to pick some standard terms here and use them repeatedly throughout the paper.
Line 137 - citation format error; also could use page number
Line 142 - same; page # helpful with quotes
Line 142 - odd to have two back-to-back quotes. no commentary at all?
Lines 160 + 161 - "its’" does not exist
Line 178 - data “were” / data is a plural noun
Line 287: This sentence is grammatically incorrect.
Line 287: colon instead of semi-colon

Tables:
I seem to be missing Table 2 in the PDF. It says “on next page” and then goes to Table 3. Is Table 3 mis-numbered? The text does not refer to a Table 2 at all.

Table 3 does not have a helpful title—it’s the “qualities” or characteristics of the included papers, instead of their “quality,” like value. It should explain the headers at least. What is being displayed? “Study Characteristics” makes more sense than “Domain” which sounds content-based.

Experimental design

Line 175 - I'm very uncertain what was done here. To what extent was there agreement among the authors? Nonagreement?
Line 182 - Who got to decide what the themes were? Was there agreement? How do you know? Can you explain?
Lines 183 - 184 - This sentence sounds like you decided what your conclusions were by talking to yourselves. Seems obvious.

Table 3:
Is Assessment by study participant not objective in nature? What is meant by this distinction?
What is meant by a fixed study design? Why are those being singled out?
Who decided the data analyses were appropriate for the study designs? Was there any disagreement about this?
What is meant by “Beyond descriptive analysis?”
The Table formatting seems odd but that may be the result of the PDF processing. The data in the third column should be shifted over to the right.

Figure 4—some of the references are not correctly formatted. Under Value, I see “botden et)

My sense is providing the first author and the year would be better than this format, but perhaps this is the standard in the field?

Validity of the findings

Line 148 - does the analysis need to be updated? the end of the data collection is now 2 years old

Was the goal of evaluating user acceptance ever evaluated? it’s mentioned as one of the 3 goals in the intro. This needs to be revisited in discussion, in any case. Ideally the discussion section should mirror and extend the introduction to help the reader understand what is going on.

The paragraph on cost seems irrelevant for purpose of paper.

Reviewer 3 ·

Basic reporting

See below

Experimental design

See below

Validity of the findings

See below

Additional comments

I've looked over the rebuttal letter and the changes and think that the authors have done a good job in making the requested changes. The manuscript has much improved and is in my view suitable for publication in PeerJ.

---

## Round 0.3 · accepted · Accept

As this revision appears to satisfy all of the reviewer comments, I believe it is suitable for publication.

I did notice several typographical errors - I suggest careful proofreading, if possible by a native English speaker.